# Dynamic CYP2E1 expression and metabolic activity changes in male rats during immune liver injury and sex differences in alcohol metabolism

**Jiayi Zhang◉, Yingying Cao◉, Ziqi Jin, Runa A, Xiaoxue Wang, Lingyu Zhang, Yingqi Hu, Yongzhi Xue◉\***

Department of Pharmacology, Baotou Medical College, Institute of Pharmacokinetics and Liver Molecular Pharmacology, Baotou, China

◉ These authors contributed equally to the work.
\* xyzhxyzh68@sohu.com, xyzhxyzh68@sohu.com

## Abstract

Previous studies conducted by our team have demonstrated that CYP2E1 expression is downregulated during Bacillus Calmette-Guérin (BCG)-induced immune liver injury (hepatitis). However, the dynamic changes in CYP2E1 metabolic activity during the acute, chronic, and recovery phases of hepatitis remain unclear. This study developed a non-invasive approach using a breath alcohol analyzer to assess CYP2E1 metabolic activity through alcohol metabolism and examined sex-based differences in alcohol metabolism in rats. Using a BCG-induced male rat hepatitis model, we investigated the dynamic changes in CYP2E1 metabolic activity at different stages of hepatitis and explored the underlying mechanisms. The results indicated that the breath alcohol analysis method exhibited high precision, linearity, and reproducibility in assessing CYP2E1 metabolic activity. CYP2E1 metabolic activity and protein expression displayed an induction trend with increased alcohol intake (P < 0.05). Female rats exhibited significantly higher CYP2E1 metabolic activity compared to males (P < 0.05), indicating significant sexual dimorphism. On day 6 post-BCG stimulation, CYP2E1 metabolic activity was most severely impaired (P < 0.05). Notably, alterations in metabolic activity were detected earlier and were more pronounced than changes in protein expression. Similar dynamic changes were observed in the hepatic NF-κB inflammatory pathway and the MAPK oxidative stress pathway. In conclusion, the breath alcohol analysis method is an effective tool for assessing CYP2E1 metabolic activity in rats. CYP2E1 can be significantly induced following a single high dose of alcohol, with female rates exhibiting greater metabolic activity compared to males. CYP2E1 metabolic activity showed the most notable impairment on day 6 post-BCG, with gradual recovery observed at days 10 and 14, and parallel changes observed in inflammatory and MAPK pathways. The recovery of CYP2E1 protein expression occurred after 14 days, which was later than that of the metabolic activity.

**Data availability statement:** All data in this manuscript have been submitted as supporting information.

**Funding:** This research was supported by the National Natural Science Foundation of China (81460567 and 82160709), and the Natural Science Foundation of Inner Mongolia Autonomous Region (2014MS0813, 2019MS08198 and 2023MS08066).

**Competing interests:** There are no conflicts of interest to declare.

## Introduction

Cytochrome P450 2E1 (CYP2E1), a critical member of the P450 enzyme family, is responsible for the oxidation and reduction of approximately 8% of chemicals, including alcohol, drugs, toxins, lipids, and carcinogens [1]. It plays a key role in the metabolism of various substances, such as inhaled anesthetics (e.g., enflurane and halothane), chlorzoxazone, acetaminophen, and ketamine hydrochloride [2]. Upon exposure to immune factors, viruses, drugs, ethanol, or other harmful exogenous substances, the liver activates immune responses, triggering severe inflammatory processes. During this process, CYP2E1 undergoes significant alterations and is actively involved in the pathophysiological mechanisms of liver injury [3]. Although BCG-induced immune stimulation has been shown to significantly suppress CYP2E1 metabolic activity and protein expression, the dynamic changes in CYP2E1 through-out the progression of liver injury remain insufficiently characterized [4,5], warranting further investigation.

The liver is the principal organ responsible for metabolizing over 90% of alcohol into acetaldehyde. At low blood alcohol concentrations, alcohol dehydrogenase (ADH) predominantly catalyzes this process. However, when blood alcohol concentrations exceed 15–20 mg%, ADH becomes saturated owing to its low Km value, and CYP2E1 assumes a dominant role in catalyzing alcohol metabolism. This process generates acetaldehyde and reactive oxygen species (ROS), contributing to oxidative stress and liver injury [6]. Therefore, the rate of alcohol metabolism can serve as an indicator of CYP2E1 metabolic activity, particularly when blood alcohol concentrations exceed 15–20 mg%. Traditional methods for evaluating alcohol metabolism involve collecting blood samples from the tail or orbital venous plexus of rats, followed by gas chromatography to determine blood alcohol levels [7]. However, these methods necessitate animal sacrifice following blood collection, precluding repeated measure-ments and dynamic assessments of CYP2E1 activity [8]. Although breath alcohol analyzers have been shown to accurately reflect blood alcohol concentrations [9,10], their application in rats is limited owing to the small lung capacity of rodents and their inability to exhale actively. To address these limitations, this study employed a rat breath collection bottle method [11], using a 550 mL syringe to collect exhaled air over a 10-minute period for subsequent analysis. This non-invasive approach allows rats to move freely during sampling, better aligning with pharmacokinetic requirements [12] and overcoming the limitations associated with gas chromatography [8]. More-over, this method facilitates repeated measurements over time, enabling the con-struction of blood alcohol concentration-time curves, and is well-suited for CYP2E1 pharmacokinetic investigations.

In this study, male and female rats were administered alcohol via gavage on days 1, 3, and 5. Alcohol concentrations were measured using the breath collection bottle method to establish and validate the methodology, while simultaneously assessing sex-based differences and alcohol-induced alterations in CYP2E1 activity. Further-more, an immune-mediated liver injury model was established through intravenous injection of Bacillus Calmette-Guérin (BCG) to investigate the dynamic changes in CYP2E1 across different stages of hepatitis and elucidate its regulatory mechanisms.

This study aims to enhance the understanding of CYP2E1 regulation and identify potential therapeutic targets and strategies for the treatment of hepatitis.

## Materials and methods

### Experimental animals and reagents

Specific pathogen-free (SPF) Wistar rats (male and female), aged 8–9 weeks and weighing $220 \pm 20$ g, were provided by Sibefu (Beijing) Biotechnology Co., Ltd. (License No.: SCXK (Jing) 2019−0010). The rats were housed under a 12/12 h light/dark cycle at a temperature of $23 \pm 2$ °C, with access to food and water ad libitum. All experimental procedures were conducted in accordance with the ARRIVE guidelines and relevant regulations and were approved by the Ethics Committee of Baotou Medical College (Approval No.: 2021−061). At the conclusion of the experiments, the rats were euthanized under anesthesia using sodium pentobarbital. The rats were euthanized by intraperitoneal injection of 5% pentobarbital sodium solution at 200 mg/kg. The rats died without spontaneous respiration, cardiac arrest, and dilated pupils, and euthanized by neck amputation when necessary.

The following reagents were used in the study: Bacillus Calmette-Guérin (BCG) vaccine (China Ruiyu Biotechnology Co., Ltd., Batch No.: 202301, Shanghai); TNF-α rabbit polyclonal antibody (No. BA0131) and HRP-goat anti-rabbit IgG (No. BA1056) were obtained from Wuhan Boster Biological Engineering Co., Ltd.; β-tubulin rabbit polyclonal antibody (No. bsm-33034M), IL-6 rabbit polyclonal antibody (No. bs-0782R), p38 rabbit polyclonal antibody (No. bs-28027R), p-p38 rabbit polyclonal antibody (No. bs-0636R), JNK rabbit polyclonal antibody (No. bs-10562R), p-JNK rabbit polyclonal antibody (No. bs-17591R), and CYP2E1 rabbit anti-rat polyclonal antibody (No. bs-4562R) were obtained from Beijing Biosynthesis Biotechnology Co., Ltd.; GAPDH mouse polyclonal antibody (No. 41549) and NF-κB p65 rabbit polyclonal antibody (No. 27518) were obtained from Shanghai Sabo Biotechnology Co., Ltd.

### Validation of the breath alcohol analyzer method

Reproducibility: A 1 mL volume of 1.75% (v/v) alcohol was placed at the bottom of a gas collection bottle. After 10 minutes, the gas was extracted, and the alcohol concentration was measured using the Black Cat 800 breath alcohol analyzer (Shenzhen Zhaowei Technology Co., Ltd.). This procedure was repeated nine times, and the coefficient of variation (CV) was calculated to assess reproducibility. Time dependency: A 1 mL volume of 1.75% (v/v) alcohol was placed at the bottom of the gas collection bottle. Alcohol concentrations were measured at 1, 2, 3, 4, and 5 minutes to plot a time-concentration curve. Concentration dependency: A 1 mL volume of alcohol at concentrations of 0.21875, 0.4375, 0.875, 1.75, and 3.5 mg/mL was placed at the bottom of the gas collection bottle. After 10 minutes, the alcohol concentration was measured, and a concentration-response curve was constructed. Limit of detection: To determine the limit of detection, a 3.5% (v/v) alcohol solution was serially diluted until the detection value reached zero.

### Measurement of CYP2E1 metabolic activity in rats

Rats were administered 5 mL/kg of 56% (v/v) alcohol via gavage and placed in a gas collection bottle. After 10 minutes, the alcohol concentration was measured using the breath alcohol analyzer. The device automatically converted the measured breath alcohol concentration to the corresponding blood alcohol concentration. After each measurement, the rats were then returned to their cages for a 10-minute rest period with free access to food and water, before being placed back into the gas collection bottle for subsequent measurements. This process was conducted at 20-minute intervals until the alcohol concentration reached zero. A blood alcohol concentration-time curve was plotted based on the measurements. Pharmacokinetic parameters, including area under the curve (AUC) from time zero to time (0–t), mean residence time (MRT), and clearance rate (CLz/F), were calculated using DAS 3.0 software.

### Sex-based differences in CYP2E1 metabolic activity

Forty rats were divided into female and male alcohol groups (n = 20 per group). On days 1, 3, and 5 of the experiment, the rats were administered 5 mL/kg of 56% alcohol via gavage. Breath alcohol concentrations were measured, and concentration-time curves were plotted to compare CYP2E1 metabolic activity between sexes.

### Dynamic changes in CYP2E1 in an immune-mediated liver injury model

Another forty male rats were randomly assigned to control, BCG 6-day, BCG 10-day, and BCG 14-day groups (n = 10 per group). Rats in the BCG groups were injected with 125 mg/kg BCG via the tail vein. CYP2E1 metabolic activity was assessed on days 6, 10, and 14 following BCG administration. Liver tissues were collected for hematoxylin and eosin (HE) staining and western blot analysis.

### Western blot analysis of CYP2E1, NF-κb, TNF-α, IL-6, p38, p-p38, JNK, and p-JNK expression

Liver tissues were weighed, and proteins were extracted according to the manufacturer's instructions. Protein concentrations were determined using the bicinchoninic acid (BCA) method. Equal amounts of protein (30 μg) were separated on 5–10% polyacrylamide gels and transferred onto polyvinylidene fluoride (PVDF) membranes. Non-specific binding sites were blocked with 5% skim milk in tris-buffered saline at 25 °C for 1 hour. The membranes were incubated using 1:1000 dilutions of rabbit polyclonal antibodies against CYP2E1, NF-κB, TNF-α, IL-6, p38, p-p38, JNK, and p-JNK at 4 °C for 12 hours, followed by incubation with 1:2000 dilutions of goat anti-rabbit IgG secondary antibody at 25 °C for 30 minutes. After washing with phosphate-buffered saline (PBS), protein bands were visualized using an enhanced chemiluminescence detection system (AR1190, Wuhan Boster Biological Engineering Co., Ltd.), and images were captured using the OmegaLum C imaging system (Gel Company, USA). GAPDH polyclonal antibody was used as the internal reference for CYP2E1, NF-κB, p38, p-p38, JNK, and p-JNK, whereas β-tubulin rabbit polyclonal antibody was used as the internal reference for TNF-α and IL-6.

### Statistical analysis

Statistical analyses were conducted using GraphPad Prism version 8.0 for Windows (GraphPad Software, La Jolla, California, USA). Following a normality test, data conforming to a normal distribution were expressed as mean ± standard deviation. For multiple comparisons, one-way analysis of variance (ANOVA) was used, while comparisons between two groups were conducted using a between-group t-test. A confidence level of 95% was applied to all analyses, with $P < 0.05$ considered statistically significant.

## Results

### Methodological validation of the breath alcohol detection method for assessing CYP2E1 metabolic activity

The results of the methodological validation are presented in Fig 1. Fig 1A illustrates the experimental setup for detecting breath alcohol concentration using the gas collection bottle method. The coefficient of variation (CV) calculated from nine replicate measurements was 12.55% (Fig 1B), demonstrating satisfactory reproducibility and minimal experimental error. The time- and concentration-dependent relationships of alcohol concentration (Figs 1C and 1D) demonstrated strong linearity, with a goodness-of-fit ($R^2$) greater than 0.95. The limit of detection for the method was 0.44% (v/v). These findings confirm that the method exhibits adequate reproducibility, precision, and a suitable linear range, rendering it suitable for subsequent breath alcohol concentration measurements.

### Sex-based differences in CYP2E1 metabolic activity and dynamic changes in protein expression

As illustrated in Fig 2A, the experimental design involved the administration of 56% alcohol (5 mL/kg) via gavage. In male rats, blood alcohol concentrations at 80, 100, 180, 220, 240, 260, 280, 300, and 340 minutes were significantly higher

 

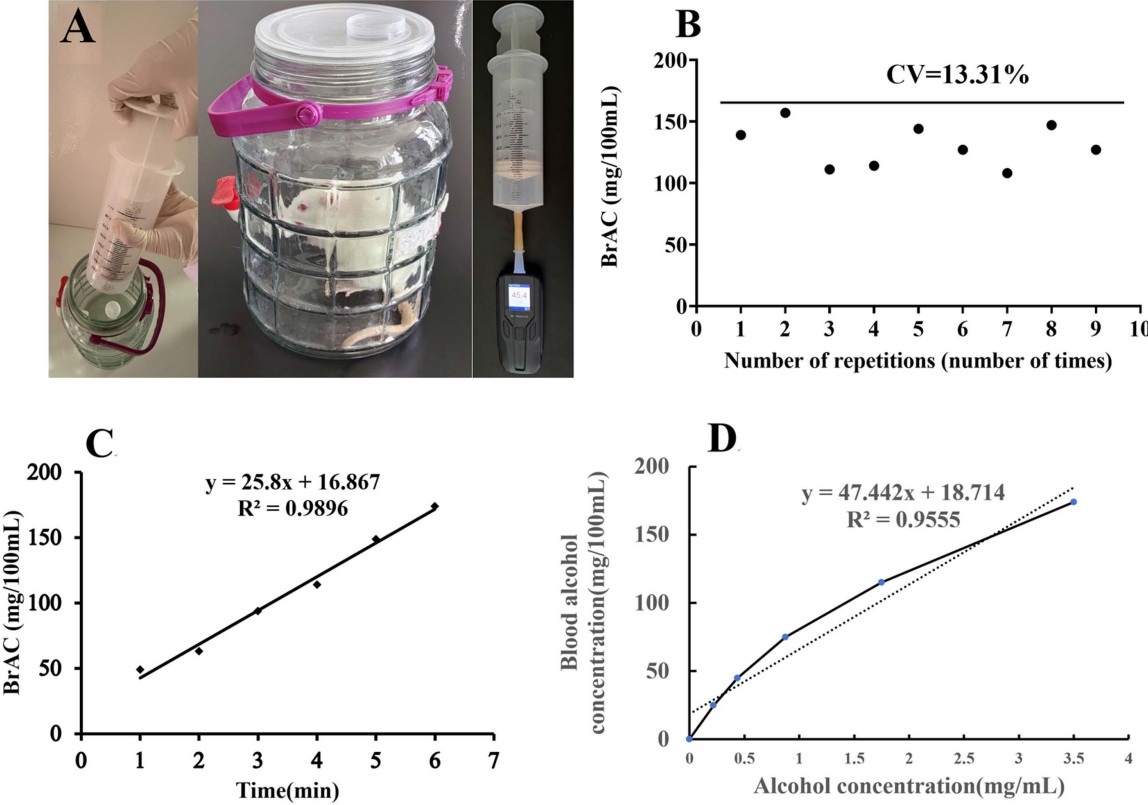

**Fig 1. Methodological validation of breath alcohol monitoring in rats.** (A) Experimental setup for breath alcohol concentration measurement using the gas collection bottle method. (B) Coefficient of variation analysis (C) Time-dependent alcohol concentration curve. (D) Concentration-dependent alcohol response curve.

following the second administration compared to the first (Fig 2B, P<0.05). After the third administration, blood alcohol concentrations at 100 and 220 minutes were also significantly lower than those observed after the first administration (Fig 2B, P<0.05). Pharmacokinetic analysis (Table 1) revealed that the AUC(0–t) was significantly reduced after the second and third administrations compared to the first (P<0.05), indicating a decrease in total blood alcohol levels. The MRT(0–t) was significantly shortened after the second administration (P<0.05), suggesting an acceleration in alcohol metabolism. Additionally, CLz increased progressively across administrations, indicating enhanced alcohol elimination, confirming that alcohol administration induces CYP2E1 metabolic activity in male rats.

In female rats, blood alcohol concentrations at 40, 80, 100, and 120 minutes were significantly lower after the third administration compared to the first (Fig 2C, P<0.05). Pharmacokinetic parameters (Table 2) revealed that the AUC(0–t) was significantly reduced after the second administration (P<0.05), indicating accelerated alcohol metabolism, whereas CLz increased, suggesting enhanced alcohol elimination. These results confirm that alcohol exposure induces CYP2E1 metabolic activity in female rats as well.

Following the first administration, blood alcohol concentrations in male rats were significantly higher than those in female rats at 75, 195, 210, and 225 minutes (Fig 2D, P<0.05), suggesting stronger CYP2E1 metabolic activity in females. However, after the second and third administrations, no significant differences in blood alcohol concentrations were observed between males and females (Figs 2E and 2F), indicating that alcohol exerts a stronger inductive effect on CYP2E1 metabolic activity in males.

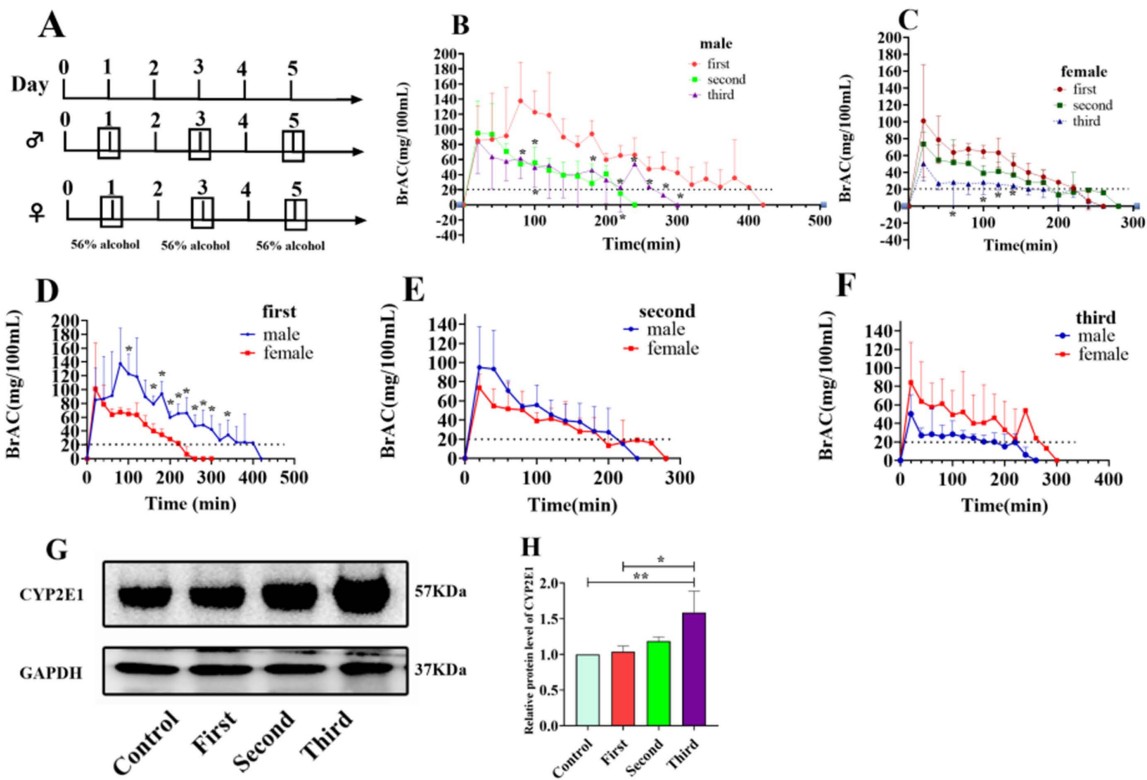

**Fig 2. Sex-based differences in alcohol metabolism and induction of CYP2E1 protein expression in male rats.** (A) Experimental design. (B) Effect of gavage frequency on blood alcohol concentration in male rats. Blood alcohol concentrations were measured using a breath alcohol analyzer following the administration of 56% alcohol (5 mL/kg) on days 1 (first administration), 3 (second administration), and 5 (third administration). *P < 0.05 compared to the first administration. (C) Effect of gavage frequency on blood alcohol concentration in female rats. *P < 0.05 compared to the first administration. (D) Effect of sex on blood alcohol concentration after the first administration. *P < 0.05 compared to female rats. (E) Effect of sex on blood alcohol concentration after the second administration. (F) Effect of sex on blood alcohol concentration after the third administration. (G) Effect of gavage frequency on CYP2E1 protein expression in rat livers. Liver proteins were extracted following alcohol gavage on days 1 (first administration), 3 (second administration), and 5 (third administration), and CYP2E1 expression was analyzed through western blotting using 30 μg of total protein per sample. GAPDH was used as the internal loading control. (H) Effect of gavage frequency on relative CYP2E1 protein expression levels. CYP2E1 protein expression levels were quantified using ImageJ software (NIH, USA). Data are presented as the mean ± SD from three independent experiments. *P < 0.05; **P < 0.01; ***P < 0.001.

**Table 1. Effect of alcohol administration frequency on pharmacokinetic parameters in male rats ($\bar{x} \pm s$, n = 3).**

| alcohol administration frequency | $AUC_{(0-t)}$/μg/(L·min) | $MRT_{(0-t)}$/min | CLz/L/(min·kg) |
|---|---|---|---|
| First | 27000.00 ± 1294.30 | 156.46 ± 34.97 | 103.87 ± 5.11 |
| Second | 11146.67 ± 4343.01* | 86.42 ± 10.39* | 262.84 ± 99.54 |
| Third | 10953.33 ± 8163.56* | 92.12 ± 27.08* | 348.22 ± 200.83 |

Compared to the first administration, *P < 0.05. AUC(0–t): Area under the blood alcohol concentration-time curve; MRT(0–t): Mean residence time; CLz/L: Clearance rate.

Western blot analysis of CYP2E1 protein expression in male rat liver tissues (Fig 2G) demonstrated uniform band densities for the internal reference protein GAPDH, confirming accurate protein quantification and consistent sample loading. Baseline CYP2E1 expression was observed in the non-gavage group (0 administrations), and CYP2E1 protein expression significantly increased with the number of alcohol gavage administrations (P < 0.05), exhibiting a clear dose-dependent

**Table 2. Effect of alcohol administration frequency on pharmacokinetic parameters in female rats (x̄±s, n=3).**

| alcohol administration frequency | AUC$_{(0-t)}$/μg/(L·min) | MRT$_{(0-t)}$/min | CLz/L/(min·kg) |
|---|---|---|---|
| First | 12386.67±2262.42 | 98.554±13.94 | 218.79±28.70 |
| Second | 8800.00±3688.96* | 92.07±15.11 | 355.56±136.94 |
| Third | 5646.67±2450.90* | 100.94±5.82 | 529.35±229.83* |

Compared to the first administration, *P<0.05. AUC(0–t): Area under the blood alcohol concentration-time curve; MRT(0–t): Mean residence time; CLz/L: Clearance rate.

trend (Figs 2G and 2H). These results confirm that alcohol administration significantly induces CYP2E1 protein expression in male rats.

## Dynamic changes in CYP2E1 expression during BCG-induced immune-mediated liver injury

**Histopathological analysis.** H&E staining results revealed normal liver tissue structure in the control group, characterized by clear hepatocyte cord arrangements and the absence of inflammatory cell infiltration (Fig 3A). In contrast, the BCG-treated groups exhibited significant pathological changes, including neutrophil and lymphocyte infiltration, disorganized hepatocyte cords, and marked congestion (Figs 3B–3D). At days 10 and 14 post-BCG treatment, the inflammatory state in the liver partially improved, with reduced inflammatory cell clusters, although congestion and edema remained evident (Figs 3C and 3D).

## Dynamic changes in CYP2E1 expression and metabolic activity

To investigate alterations in CYP2E1 expression and metabolic activity during BCG-induced immune-mediated liver injury, 56% alcohol was used as a probe and breath alcohol concentration was measured on days 6, 10, and 14 following BCG treatment. The experimental design is illustrated in Fig 4A. The results revealed that, compared to the control group, CYP2E1 metabolic activity was significantly reduced in the BCG (6, 10, and 14 d) groups, as evidenced by reduced alcohol metabolism and increased blood alcohol concentrations. Notably, the BCG (6 d) group exhibited significantly higher blood alcohol concentrations at 360 minutes compared to the BCG (14 d) group (Fig 4B, P<0.05). Pharmacokinetic analysis (Fig 4C) revealed that the AUC(0–t) was significantly increased in the BCG (6, 10, and 14 d) groups (P<0.01) compared to the control group, indicating reduced alcohol metabolism and increased alcohol accumulation. Furthermore, AUC(0–t) values in the BCG 10-day and 14-day groups were significantly lower than those in the BCG 6-day group (P<0.05), suggesting a partial recovery of alcohol metabolism and reduced total alcohol levels over time. Using a blood alcohol concentration of 20 mg/100 mL as the dividing line, the area under the blood alcohol concentration curve was divided into two. The >20 mg/100 mL part reflects the metabolic activity of CYP2E1, whereas the <20 mg/100 mL reflects the metabolic activity of ADH. These findings indicate that liver injury was the most severe on day 6 post-BCG administration, corresponding to the lowest CYP2E1 metabolic activity. At days 10 and 14, metabolic activity exhibited partial recovery; however, it remained significantly lower than that in the control group (P<0.05). On the 6th day after BCG vaccine administration, the corresponding ADH metabolic activity was also the lowest (P<0.05).

Western blot analysis of CYP2E1 protein expression in the liver tissues of the BCG-treated groups (6, 10, and 14 d) demonstrated a significant downregulation of CYP2E1 expression on day 6 compared to the control group (P<0.01), representing the lowest expression level observed. Although CYP2E1 expression increased slightly at days 10 and 14, the change was not statistically significant, indicating only partial recovery (Fig 4D).

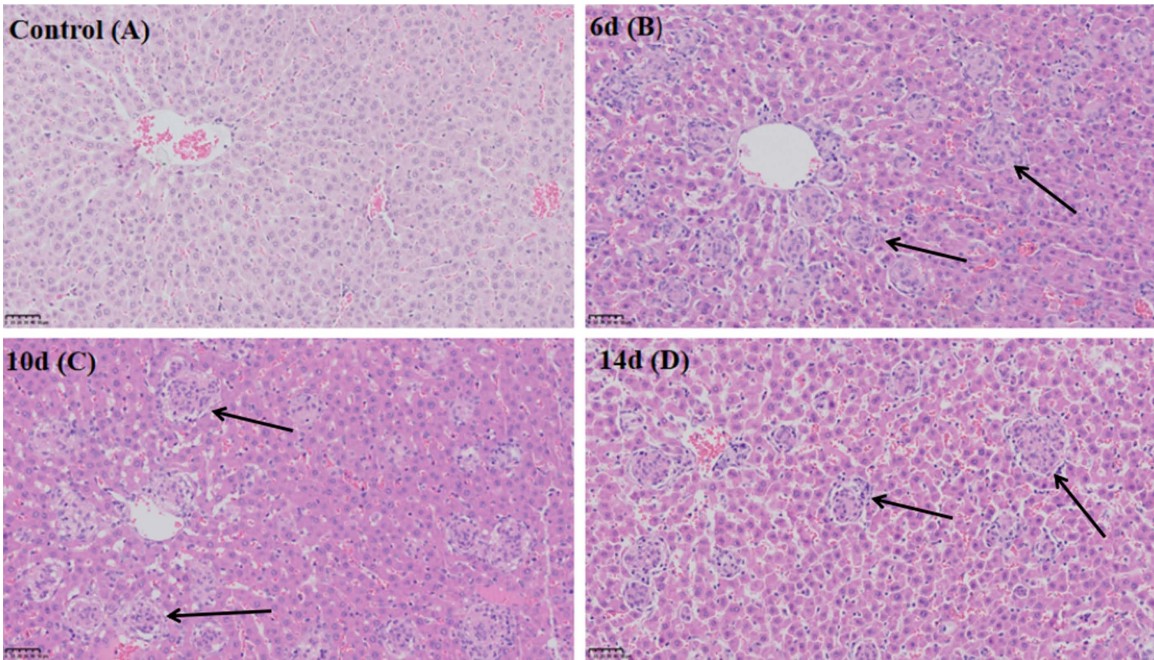

**Fig 3. Histopathological changes in rat livers at different time points following BCG treatment (H&E, 200X).** Scale bar: 50 μm. (A) Control group. (B) Liver tissue on day 6 following BCG treatment. (C) Liver tissue on day 10 following BCG treatment. (D) Liver tissue on day 14 following BCG treatment. Arrows indicate inflammatory cell clusters. BCG: Bacillus Calmette-Guérin; H&E: Hematoxylin and eosin.

### Dynamic changes in inflammatory cytokines (NF-κB, TNF-α, IL-6) and MAPK-related factors (p38, p-p38, JNK, p-JNK) during BCG-induced immune-mediated liver injury

Western blot analysis revealed dynamic changes in the expression of inflammatory cytokines in rat livers on days 6, 10, and 14 following BCG treatment. Compared to the control group, the expression levels of the pro-inflammatory cytokines NF-κB, TNF-α, and IL-6 were significantly elevated on day 6 ($P < 0.01$), reaching peak levels, indicating the most severe stage of liver injury. Over time, the expression levels of these cytokines gradually decreased; however, the differences were not statistically significant (Figs 5A–5E).

Further analysis of MAPK pathway-related protein expression and phosphorylation levels (Figs 5F–5H) revealed that, compared to the control group, the ratios of p-p38/p38 and p-JNK/JNK were significantly elevated on day 6 ($P < 0.05$), reaching their peak at this time point. Notably, p-p38 levels remained significantly elevated on day 10 ($P < 0.05$), whereas p-JNK expression peaked on day 6. These findings suggest that the activation of the p38 and JNK signaling pathways in response to oxidative stress may occur asynchronously. As the liver tissue underwent self-repair, the expression levels of p38, p-p38, JNK, and p-JNK gradually decreased, indicating a gradual alleviation of the inflammatory response.

## Discussion

This study elucidates the time-dependent changes and sex-based differences in CYP2E1 metabolic activity in rats. Repeated alcohol administration significantly induced CYP2E1 activity after three days. Male rats exhibited a 2.4-fold increase in CYP2E1 activity following the second gavage, with no further significant change after the third. In contrast, female rats demonstrated a 1.4-fold increase after the second gavage, with no significant induction after the third, indicating a stronger induction effect in males and a more pronounced response following the second gavage. However, female rats consistently demonstrated faster alcohol metabolism, which may be attributed to sex-specific differences in CYP2E1

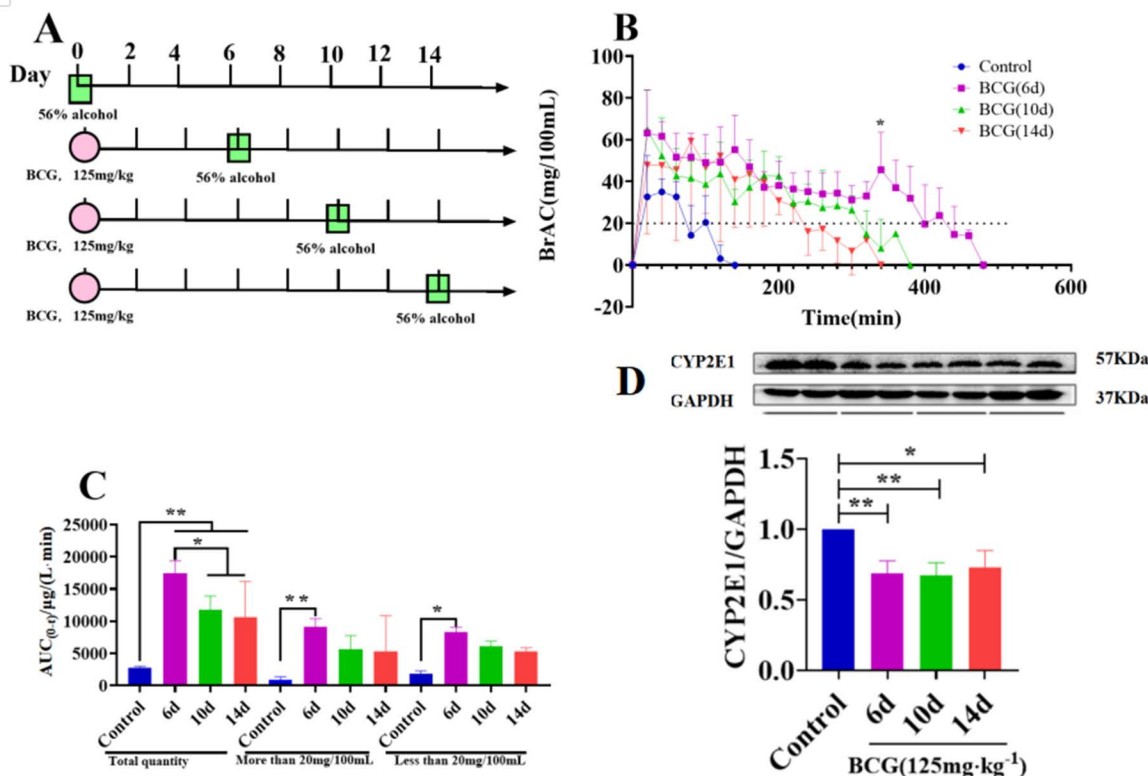

**Fig 4. Dynamic changes in CYP2E1 expression during immune-mediated liver injury.** Rats were intravenously injected with BCG (125 mg/kg) and administered 56% alcohol (5 mL/kg) via gavage on days 6, 10, and 14. Blood alcohol concentrations were measured using a breath alcohol analyzer. (A) Experimental design. (B) Time-dependent changes in alcohol metabolism on days 6, 10, and 14 following BCG treatment. *$P < 0.05$ compared to the control group. (C) Effect of BCG treatment on AUC(0–t) on days 6, 10, and 14. *$P < 0.05$; **$P < 0.01$; ***$P < 0.001$. Liver proteins were extracted, and CYP2E1 expression was analyzed through western blotting using 30 μg of protein. GAPDH was used as a loading control. CYP2E1 expression levels were quantified using ImageJ software (1.46, NIH, USA). Data are presented as the mean ± SD from three independent experiments. (D) Changes in CYP2E1 protein expression on days 6, 10, and 14 following BCG treatment. *$P < 0.05$; **$P < 0.01$. BCG: Bacillus Calmette–Guérin.

expression. Higher baseline hepatic CYP2E1 protein levels, combined with increased oxidative stress and greater susceptibility to liver damage following ethanol exposure, may account for the reduced induction of CYP2E1 activity in females [13]. Additionally, fluctuations in sex hormones, differences in hepatic sex hormone receptor expression, and variations in growth hormone secretion patterns may further contribute to these sex-based metabolic differences [14]. Notably, estrogen has been reported to enhance alcohol metabolism [14], which aligns with the findings of the present study. Unlike rats, human males generally exhibit faster ethanol metabolism compared to females [15]. Furthermore, chronic alcohol consumption in humans can result in the development of tolerance and accelerated alcohol clearance [16]. Our findings demonstrate that repeated alcohol exposure upregulates CYP2E1 expression and activity in rats, with a more robust induction observed in males. A single high-dose alcohol exposure significantly induced CYP2E1 activity, whereas protein expression exhibited a dose-dependent increase following three gavages. These findings suggest that acute alcohol exposure may enhance the metabolic elimination of certain drugs, potentially leading to therapeutic failure. Conversely, chronic alcohol consumption may upregulate CYP2E1 expression, contributing to oxidative stress and liver injury. Therefore, limiting alcohol consumption is particularly advisable for individuals undergoing pharmacological treatment.

In BCG-induced liver inflammation, CYP2E1 expression, along with NF-κB, and MAPK pathway activity, exhibited dynamic changes. CYP2E1 activity decreased following BCG administration, with the lowest protein expression observed

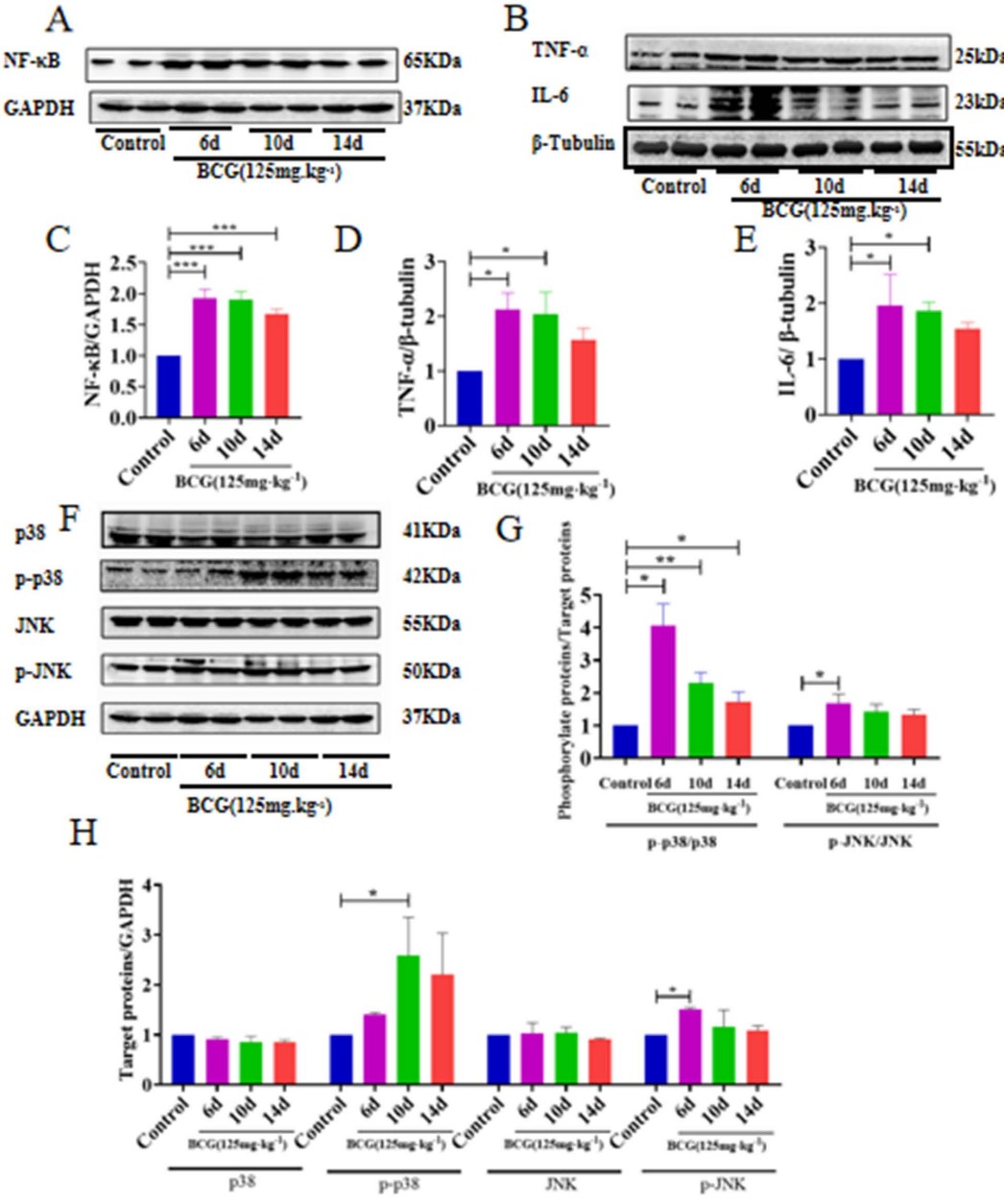

**Fig 5. Dynamic changes in inflammatory cytokines and MAPK-related factors during BCG-induced immune-mediated liver injury.** Rats were intravenously injected with BCG (125 mg/kg), and liver proteins were extracted on days 6, 10, and 14 to evaluate the expression levels of inflammatory cytokines (NF-κB, TNF-α, IL-6) and MAPK pathway-related factors (p38, p-p38, JNK, p-JNK). Western blot analysis was performed using 30 µg of protein, with results normalized to GAPDH or β-tubulin. Protein expression levels were quantified using ImageJ software (NIH, USA). Data are presented as mean ± SD from three independent experiments. (A, C) NF-κB; (B, D, E) TNF-α, IL-6; (F, G, H) p38, p-p38, JNK, and p-JNK protein expression levels. *P < 0.05; **P < 0.01; ***P < 0.001. BCG: Bacillus Calmette-Guérin.

on day 6. The discrepancy between metabolic activity and protein expression may be attributable to post-translational modifications, such as SUMOylation, which stabilizes CYP2E1 and prolongs its half-life [16,17]. Ethanol binding may also enhance CYP2E1 stability [16], which could explain the more pronounced changes in activity compared to protein

expression. CYP2E1 activity began to recover on day 10, whereas protein expression recovery was delayed after day 14, possibly owing to the regulatory influence of the MAPK and NF-κB pathways. NF-κB activation, driven by IL-6 and TNF-α, exacerbates inflammatory responses [18], whereas MAPK pathway activation by ROS triggers mitochondrial dysfunction [19,20]. These pathways amplify inflammatory responses, leading to delayed CYP2E1 recovery [21,22]. During acute hepatitis, CYP2E1 activity is significantly impaired, with a slower recovery of protein expression, highlighting the importance of restoring hepatic metabolic function as part of hepatitis treatment. However, this study has certain limitations. Breath alcohol testing may not exclusively reflect CYP2E1 activity, as ADH are also involved in alcohol metabolism. When the blood alcohol concentration is below 15–20 mg%, ADH mainly catalyzes this process. However, when blood alcohol concentration exceeds this value and ADH is saturated, CYP2E1 mainly catalyzes alcohol metabolism. Most of the blood alcohol concentrations in this study exceeded this critical value; consequently, alcohol was mainly metabolized by CYP2E1. The blood alcohol concentration curve was divided into two parts (using 20 mg% as threshold). The upper part reflected the metabolic activity of CYP2E1, whereas the lower part reflected the metabolic activity of ADH, which solved this problem well. Our previous experiments used the hydroxyl metabolism of clozoxazone as a probe of the metabolic activity of CYP2E1, which confirmed that the metabolic activity of CYP2E1 was decreased in BCG-induced hepatitis [3–5], which was the same as the results of this study, and also confirmed the feasibility of this method. Therefore, considering the advantages of non-invasive, simple, and multi-point field detection, this method presents a new method to replace liquid chromatography with clozoxazone as a probe for CYP2E1 metabolic activity; however, it requires high blood alcohol concentration.

## Conclusion

This study demonstrates that the proposed method is suitable for dynamically assessing blood alcohol concentration in rats, and is suitable for the analysis of CYP2E1 metabolic activity. Alcohol administration induces CYP2E1 protein expression in rats in a dose-dependent manner. Female rats exhibited significantly higher CYP2E1 metabolic activity compared to male rats. The metabolic activity of CYP2E1 was the most impaired on day 6 following BCG administration, and gradually recovered at days 10 and 14 following BCG administration. Notably, changes in metabolic activity were more pronounced than those in protein expression. Furthermore, CYP2E1 expression and activity are regulated by both MAPK and NF-κB signaling pathways. The findings imply that targeting MAPK or NF-κB signaling could indirectly regulate CYP2E1 activity, in turn opening avenues for therapies aimed at mitigating CYP2E1-mediated toxicity or improving drug efficacy in patients with dysregulated metabolic pathways.

## Supporting information

**S1 Data. Data.**
(XLSX)

## Acknowledgments

The authors thank Prof. Guo-liang Zhang of the Department of Pharmacology, Basic Medical School, Beijing University, Beijing, China, for providing technical assistance.

## Author contributions

**Conceptualization:** Jiayi Zhang, yongzhi xue.

**Data curation:** Jiayi Zhang, Yingying Cao, Ziqi Jin, Runa A, Yingqi Hu, yongzhi xue.

**Formal analysis:** Jiayi Zhang, Yingying Cao, Runa A, Lingyu Zhang, Yingqi Hu, yongzhi xue.

**Funding acquisition:** yongzhi xue.

**Investigation:** Jiayi Zhang, Runa A, Xiaoxue Wang, Lingyu Zhang, yongzhi xue.

**Methodology:** Jiayi Zhang, Yingying Cao, Ziqi Jin, Xiaoxue Wang, Lingyu Zhang, Yingqi Hu, yongzhi xue.

**Project administration:** Yingying Cao, yongzhi xue.

**Resources:** Jiayi Zhang, Yingying Cao, Yingqi Hu, yongzhi xue.

**Software:** Jiayi Zhang, Yingying Cao, Ziqi Jin, Runa A, Yingqi Hu, yongzhi xue.

**Supervision:** yongzhi xue.

**Validation:** Jiayi Zhang, Ziqi Jin, Xiaoxue Wang, yongzhi xue.

**Visualization:** Jiayi Zhang, Yingqi Hu, yongzhi xue.

**Writing – original draft:** Jiayi Zhang, Yingying Cao, yongzhi xue.

**Writing – review & editing:** Jiayi Zhang, yongzhi xue.

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
