## [Decision Letter · Decision Letter 0]

May 25 2025

PONE-D-25-09524Dynamic alterations in CYP2E1 expression and metabolic activity and sex-based variations in alcohol metabolism during immune liver injuryPLOS ONE

Dear Dr. xue,

Thank you for submitting your manuscript to PLOS ONE. After careful consideration, we feel that it has merit but does not fully meet PLOS ONE’s publication criteria as it currently stands. Therefore, we invite you to submit a revised version of the manuscript that addresses the points raised by the reviewer.

We look forward to receiving your revised manuscript.

Kind regards,

Matias A Avila, Ph.D.

Academic Editor

PLOS ONE

2. To comply with PLOS ONE submissions requirements, in your Methods section, please provide additional information regarding the experiments involving animals and ensure you have included details on (1) methods of sacrifice (concentration of sodium pentobarbital), and (2) efforts to alleviate suffering.

Additional Editor Comments (if provided):

Reviewers' comments:

Reviewer's Responses to Questions

**Comments to the Author**

1. Is the manuscript technically sound, and do the data support the conclusions?

Reviewer #1: Partly

2. Has the statistical analysis been performed appropriately and rigorously? 

Reviewer #1: I Don't Know

3. Have the authors made all data underlying the findings in their manuscript fully available?

Reviewer #1: Yes

4. Is the manuscript presented in an intelligible fashion and written in standard English?

Reviewer #1: Yes

5. Review Comments to the Author

Reviewer #1: This is an interesting study investigating alcohol metabolism in two experimental models: one assessing the effects of repeated oral alcohol administration, including sex differences, and another examining alcohol metabolism during different phases of a BCG-induced hepatitis model. The authors developed a non-invasive breath alcohol analysis method (previously developed by the group) to assess alcohol metabolism and demonstrate its accuracy and reproducibility. The study highlights sex-based differences, showing that female rats exhibited a higher alcohol metabolism rate than males. On the other hand, when studying how alcohol metabolism is modulated in the context of hepatitis, the authors report that alcohol clearance was significantly impaired on day 6 post-BCG, which they associated with increased inflammation and activation of the p38 and JNK pathways. A gradual recovery of metabolic activity was reported by days 10 and 14 after BCG administration. These results were only tested in males.

While the study is original, there are many points that should be addressed to strengthen the conclusions and improve the manuscript. Figures require a lot of editing work. The figures contain different font types and sizes. The formatting should be reviewed to ensure consistency. The manuscript should also be extensively revised as it contains many incorrect, inaccurate, and often misleading statements, some of which are highlighted below in specific points:

Major comments:

-The specific statistical test used is not mentioned in any instance. This information is essential for the acceptance of the publication.

- Please clarify... In line 193, it states that after the third administration, blood alcohol concentration was higher than after the first administration. However, this statement is incorrect, as the data show that alcohol concentration is actually lower after the third administration compared to the first. A similar correction applies to line 203.

-Given that the blood alcohol concentration in females is slightly lower (below 100 mg/dL), I wonder whether ADH plays a more significant role, rather than CYP2E1 activity.

-I wonder how hepatic CYP2E1 expression is in females. Are there differences that could explain the different dynamics of blood alcohol levels between females and males?

- The authors state that the inflammatory state in the liver improved, suggesting a resolution of inflammation. However, they also state that congestion and edema remained evident, which are key indicators of ongoing inflammation and tissue damage. I do not see clear evidence of tissue improvement. This claim should be specifically investigated to be properly supported.

- Figure 4 presents the results of the injury model induced by BCG administration. The experimental design indicates that a control group without BCG administration was included, and blood alcohol levels were measured. How can such a different curve from that in Figure 2 be explained? I am specifically referring to the values after the first administration with 56% alcohol.

- I have some considerations regarding the Western blots presented in Figure 5. I find it noteworthy that the banding pattern observed for GAPDH is very similar to that seen for β-tubulin. Additionally, the uncropped β-tubulin image does not match the one shown in Figure 5B.

- In the discussion, some statements should be revised as they are not fully supported by the results obtained. For example, in line 371, it is stated that the minimum CYP2E1 expression is observed at 10 days post-BCG administration. However, the results show that its expression is already low by day 6. Another unclear statement in the discussion is: 'CYP2E1 activity began to recover on day 6, whereas protein expression recovery was delayed until day 10...' However, Figure 4D shows that on day 14, protein expression remains low, similar to day 10.

-As the authors state in the final part of the discussion, the reduction in alcohol levels is not always directly related to increased CYP2E1 activity, as alternative metabolic pathways may also contribute. Therefore, I suggest reviewing the manuscript to clarify these statements. If the authors intend to discuss CYP2E1 activity, I recommend including a direct assay, such as chlorzoxazone hydroxylation, to confirm its increase. Additionally, it would be useful to rule out the influence of ADH activity

- I recommend modifying the title of the paper as it does not fully represent the results of the study. The current title, 'Dynamic alterations in CYP2E1 expression and metabolic activity and sex-based variations in alcohol metabolism during immune liver injury,' suggests a broader scope. However, sex-based variations were only studied in the model of repeated high-dose alcohol administration. In the immune-mediated liver injury model, the dynamic changes in expression were studied exclusively in males.

Minor comments:

- Figure 1D presents the results of the concentration-dependent measurements. However, I do not understand why the authors refer to 1, 2, 3, 4, and 5 mg/mL, as different values are indicated in the Methods section (line 118).

- I suggest using the same scale on the x-axis for Figures 2B and 2C, as this will better highlight the differences between males and females

- Line 262. “By” implies something happened before and continued until those days, whereas "at" better indicates specific time points of measurement.

- Figures contain different font types and sizes. The formatting should be reviewed to ensure consistency.

6. PLOS authors have the option to publish the peer review history of their article (what does this mean? ). If published, this will include your full peer review and any attached files.

**Do you want your identity to be public for this peer review?** For information about this choice, including consent withdrawal, please see our Privacy Policy .

Reviewer #1: No

---

## [Author Response · Author response to Decision Letter 1]

23 Apr 2025

Reply: Modifications have been made.

2. To comply with PLOS ONE submissions requirements, in your Methods section, please provide additional information regarding the experiments involving animals and ensure you have included details on (1) methods of sacrifice (concentration of sodium pentobarbital), and (2) efforts to alleviate suffering.

Reply: Modifications have been made. The rats were euthanized by intraperitoneal injection of 5% pentobarbital sodium solution at 200 mg/kg. The rats died without spontaneous respiration, cardiac arrest, and dilated pupils, and euthanized by neck amputation when necessary.

Reply: The blotting/gel image data are fully included in the submitted paper and supplementary documents.

Additional Editor Comments (if provided):

Reviewers' comments:

Reviewer's Responses to Questions

Comments to the Author

1. Is the manuscript technically sound, and do the data support the conclusions?

Reviewer #1: Partly

2. Has the statistical analysis been performed appropriately and rigorously?

Reviewer #1: I Don't Know

3. Have the authors made all data underlying the findings in their manuscript fully available?

Reviewer #1: Yes

4. Is the manuscript presented in an intelligible fashion and written in standard English?

Reviewer #1: Yes

5. Review Comments to the Author

Reviewer #1: This is an interesting study investigating alcohol metabolism in two experimental models: one assessing the effects of repeated oral alcohol administration, including sex differences, and another examining alcohol metabolism during different phases of a BCG-induced hepatitis model. The authors developed a non-invasive breath alcohol analysis method (previously developed by the group) to assess alcohol metabolism and demonstrate its accuracy and reproducibility. The study highlights sex-based differences, showing that female rats exhibited a higher alcohol metabolism rate than males. On the other hand, when studying how alcohol metabolism is modulated in the context of hepatitis, the authors report that alcohol clearance was significantly impaired on day 6 post-BCG, which they associated with increased inflammation and activation of the p38 and JNK pathways. A gradual recovery of metabolic activity was reported by days 10 and 14 after BCG administration. These results were only tested in males.

While the study is original, there are many points that should be addressed to strengthen the conclusions and improve the manuscript. Figures require a lot of editing work. The figures contain different font types and sizes. The formatting should be reviewed to ensure consistency. The manuscript should also be extensively revised as it contains many incorrect, inaccurate, and often misleading statements, some of which are highlighted below in specific points:

Reply: The font types and sizes in the graphics have been revised to achieve uniformity. Extensive revisions were made to the manuscript.

Major comments:

-The specific statistical test used is not mentioned in any instance. This information is essential for the acceptance of the publication.

Reply: Statistical methods have been supplemented. Following a normality test, data conforming to a normal distribution were expressed as mean ± standard deviation. For multiple comparisons, one-way analysis of variance (ANOVA) was used, while comparisons between two groups were conducted using a between-group t-test.

- Please clarify... In line 193, it states that after the third administration, blood alcohol concentration was higher than after the first administration. However, this statement is incorrect, as the data show that alcohol concentration is actually lower after the third administration compared to the first. A similar correction applies to line 203.

Reply: It has been modified to have a blood alcohol concentration lower than that after the first medication.

-Given that the blood alcohol concentration in females is slightly lower (below 100 mg/dL), I wonder whether ADH plays a more significant role, rather than CYP2E1 activity.

Reply: The metabolism of alcohol in the body is mainly accomplished through the alcohol dehydrogenase (ADH) and cytochrome P4502E1 (CYP2E1) pathways [1-3]. Due to the low Km value of ADH, once the blood alcohol concentration is greater than 15-20 mg·100 mL-1, this enzyme will reach a saturated state [4]. The Km value of CYP2E1 for alcohol metabolism is relatively high (about 10 mM;) 46 mg·100 mL-1), when exposed to high concentration alcohol, its activity was significantly enhanced and it became the main metabolic enzyme [4].

-I wonder how hepatic CYP2E1 expression is in females. Are there differences that could explain the different dynamics of blood alcohol levels between females and males?

Reply: This study found that the alcohol concentration in female rats was lower than that in male rats. Female rats metabolized more alcohol, but there was no significant difference in the expression of CYP2E1 protein. The research by Li et al. indicated that, compared with female mice, the expression of CYP2E1 in male mice after alcohol treatment was lower than that in female mice. Another mechanism may be related to the fluctuations of rat gonadal hormones and the gender differences in the expression of sex hormone receptors in the liver. The estrogen secreted by female rats may promote the metabolism of alcohol.

[1]Li SQ, Wang P, Wang DM, Lu HJ, Li RF, Duan LX, Zhu S, Wang SL, Zhang YY, Wang YL. Molecular mechanism for the influence of gender dimorphism on alcoholic liver injury in mice. Hum Exp Toxicol. 2019 Jan; 38 (1) : 65-81.

[2]Finn DA. The Endocrine System and Alcohol Drinking in Females. Alcohol Res. 2020 Jul 23; (2) : 40 02.

- The authors state that the inflammatory state in the liver improved, suggesting a resolution of inflammation. However, they also state that congestion and edema remained evident, which are key indicators of ongoing inflammation and tissue damage. I do not see clear evidence of tissue improvement. This claim should be specifically investigated to be properly supported.

Reply: The text has been modified to avoid misunderstandings. In the process of hepatitis induced by BCG, on the one hand, inflammation is induced, and the body is also in the process of anti-inflammatory recovery, presenting two contradictory sides. As time went by, the inflammation partially eased, but it was still very serious at 14 days. Although on the 6th day of the experiment, the hepatitis condition was severe and the metabolic activity of CYP2E1 was the lowest, in order to ensure that the state of the rats could tolerate the administration of probe drugs and intervention drugs, in our previous studies, we usually selected the rat model of BCG-induced hepatitis to measure the metabolic activity on the 14th day of the experiment.

- Figure 4 presents the results of the injury model induced by BCG administration. The experimental design indicates that a control group without BCG administration was included, and blood alcohol levels were measured. How can such a different curve from that in Figure 2 be explained? I am specifically referring to the values after the first administration with 56% alcohol.

Reply: fig4 and fig2 are two different experiments, using two batches of rats. The metabolic activity of CYP2E1 varies greatly and is influenced by many factors, such as diet and environment. Even for a group of mice in the same experiment, the data dispersion is relatively large. To reduce errors, we adopted the method of taking one from each group for simultaneous measurement to minimize the influence of the environment on the results.

- I have some considerations regarding the Western blots presented in Figure 5. I find it noteworthy that the banding pattern observed for GAPDH is very similar to that seen for β-tubulin. Additionally, the uncropped β-tubulin image does not match the one shown in Figure 5B.

Reply: Verification and revision have been carried out.

- In the discussion, some statements should be revised as they are not fully supported by the results obtained. For example, in line 371, it is stated that the minimum CYP2E1 expression is observed at 10 days post-BCG administration. However, the results show that its expression is already low by day 6. Another unclear statement in the discussion is: 'CYP2E1 activity began to recover on day 6, whereas protein expression recovery was delayed until day 10...' However, Figure 4D shows that on day 14, protein expression remains low, similar to day 10.

Reply: It has been revised and matches the result. In BCG-induced liver inflammation, CYP2E1 expression, along with NF-κB, and MAPK pathway activity, exhibited dynamic changes. CYP2E1 activity decreased following BCG administration, with the lowest protein expression observed on day 6. The discrepancy between metabolic activity and protein expression may be attributable to post-translational modifications, such as SUMOylation, which stabilizes CYP2E1 and prolongs its half-life [16, 17]. Ethanol binding may also enhance CYP2E1 stability [16], which could explain the more pronounced changes in activity compared to protein expression. CYP2E1 activity began to recover on day 10, whereas protein expression recovery was delayed after day 14, possibly owing to the regulatory influence of the MAPK and NF-κB pathways.

-As the authors state in the final part of the discussion, the reduction in alcohol levels is not always directly related to increased CYP2E1 activity, as alternative metabolic pathways may also contribute. Therefore, I suggest reviewing the manuscript to clarify these statements. If the authors intend to discuss CYP2E1 activity, I recommend including a direct assay, such as chlorzoxazone hydroxylation, to confirm its increase. Additionally, it would be useful to rule out the influence of ADH activity

Reply: reath alcohol testing may not exclusively reflect CYP2E1 activity, as ADH are also involved in alcohol metabolism. When the blood alcohol concentration is below 15–20 mg%, ADH mainly catalyzes this process. However, when blood alcohol concentration exceeds this value and ADH is saturated, CYP2E1 mainly catalyzes alcohol metabolism. Most of the blood alcohol concentrations in this study exceeded this critical value; consequently, alcohol was mainly metabolized by CYP2E1. The blood alcohol concentration curve was divided into two parts (using 20 mg% as threshold). The upper part reflected the metabolic activity of CYP2E1, whereas the lower part reflected the metabolic activity of ADH, which solved this problem well. Our previous experiments used the hydroxyl metabolism of clozoxazone as a probe of the metabolic activity of CYP2E1, which confirmed that the metabolic activity of CYP2E1 was decreased in BCG-induced hepatitis [3-5], which was the same as the results of this study, and also confirmed the feasibility of this method. Therefore, considering the advantages of non-invasive, simple, and multi-point field detection, this method presents a new method to replace liquid chromatography with clozoxazone as a probe for CYP2E1 metabolic activity; however, it requires high blood alcohol concentration.

- I recommend modifying the title of the paper as it does not fully represent the results of the study. The current title, 'Dynamic alterations in CYP2E1 expression and metabolic activity and sex-based variations in alcohol metabolism during immune liver injury,' suggests a broader scope. However, sex-based variations were only studied in the model of repeated high-dose alcohol administration. In the immune-mediated liver injury model, the dynamic changes in expression were studied exclusively in males.

Reply: Dynamic CYP2E1 expression and metabolic activity changes in male rats during immune liver injury and sex differences in alcohol metabolism

Minor comments:

- Figure 1D presents the results of the concentration-dependent measurements. However, I do not understand why the authors refer to 1, 2, 3, 4, and 5 mg/mL, as different values are indicated in the Methods section (line 118).

Reply: The manuscript Figure 1C is time-dependent. In the methodology, it is mentioned that 1, 2, 3, 4 and 5 minutes refer to observing the different times when 1 milliliter of alcohol is placed in the gas collection bottle, observing the changes in the measured values, and judging the linear relationship based on the regression equation and R2 to analyze the precision. In Figure 1D, alcohol with different gradient concentrations was placed in the gas collection bottles for 10 minutes respectively. The magnitudes of the measured values were observed. Based on the regression equation and R2, the linear relationship was judged and the precision was analyzed. Both the text and the pictures have been reviewed and there are no problems.

- I suggest using the same scale on the x-axis for Figures 2B and 2C, as this will better highlight the differences between males and females

Reply: It has been revised.

- Line 262. “By” implies something happened before and continued until those days, whereas "at" better indicates specific time points of measurement.

- Figures contain different font types and sizes. The formatting should be reviewed to ensure consis

---

## [Decision Letter · Decision Letter 1]

Jun 19 2025

PONE-D-25-09524R1Dynamic CYP2E1 expression and metabolic activity changes in male rats during immune liver injury and sex differences in alcohol metabolismPLOS ONE

Dear Dr. xue,

Thank you for submitting your manuscript to PLOS ONE. After careful consideration, we feel that it has merit but does not fully meet PLOS ONE’s publication criteria as it currently stands. Therefore, we invite you to submit a revised version of the manuscript that addresses the minor revisions indicated by the reviewer.

We look forward to receiving your revised manuscript.

Kind regards,

Matias A Avila, Ph.D.

Academic Editor

PLOS ONE

Journal Requirements:

Reviewers' comments:

Reviewer's Responses to Questions

**Comments to the Author**

1. If the authors have adequately addressed your comments raised in a previous round of review and you feel that this manuscript is now acceptable for publication, you may indicate that here to bypass the “Comments to the Author” section, enter your conflict of interest statement in the “Confidential to Editor” section, and submit your "Accept" recommendation.

Reviewer #1: All comments have been addressed

2. Is the manuscript technically sound, and do the data support the conclusions?

Reviewer #1: Yes

3. Has the statistical analysis been performed appropriately and rigorously? 

Reviewer #1: Yes

4. Have the authors made all data underlying the findings in their manuscript fully available?

Reviewer #1: Yes

5. Is the manuscript presented in an intelligible fashion and written in standard English?

Reviewer #1: Yes

6. Review Comments to the Author

Reviewer #1: The manuscript has improved significantly; the errors have been corrected and the results clarified. Moreover, the authors have properly addressed and discussed each of the comments raised during the first round of review. For the work to be suitable for publication, I have a few comments regarding the figures.

-As requested, the panel corresponding to β-tubulin (Figure 5B) has been replaced. Please ensure that the panel uses the same format and style for the border/frame in order to maintain consistency and visual coherence across the figure. Also related to the Western blots, it appears that the bands identified as p38 (41 kDa) in the main figures may not correspond to the marked ones on the uncropped membranes shown in the Supporting Material. I consider it essential for publication that this point be carefully reviewed. I wonder which bands were used for the quantification, especially considering that the intensity pattern of the bands indicated on the uncropped membranes appears to differ from those shown in the paper. For example, for p38, the pattern does not seem to vary between the control and 6d groups in the main figure, while in the uncropped membranes, the control lanes appear notably fainter than those of the 6d group.

-Figure 4: In this version, the authors have modified the graph (4C) in response to the first review request; however, the colors used in the bars do not match those in graph B, which causes confusion. Graph D should also follow the same logic.

-Figure 5: Graphs G and H do not follow the same color reference used in the previous figures. Please edit the figures to ensure consistency and make them self-explanatory.

Additionally, I believe one important point raised by the authors in their response should be explicitly mentioned in the Materials and Methods section. Specifically, authors stated:

"Reply: Fig. 4 and Fig. 2 are two different experiments, using two batches of rats. The metabolic activity of CYP2E1 varies greatly and is influenced by many factors, such as diet and environment. Even for a group of mice in the same experiment, the data dispersion is relatively large. To reduce errors, we adopted the method of taking one from each group for simultaneous measurement to minimize the influence of the environment on the results."

I consider it important to include this clarification in the Materials and Methods section so that readers do not have the same doubt and to further strengthen the interpretation of the presented results.

7. PLOS authors have the option to publish the peer review history of their article (what does this mean? ). If published, this will include your full peer review and any attached files.

**Do you want your identity to be public for this peer review?** For information about this choice, including consent withdrawal, please see our Privacy Policy .

Reviewer #1: No

---

## [Author Response · Author response to Decision Letter 2]

6 May 2025

Reply:

1.The manuscript has been revised to ensure consistency between the p38 band (41 kDa) in Figure 5 of the paper and the corresponding band marked on the uncut film presented in the supporting information. The p38 strip in the paper is accurate; however, the position of the box in the supporting material was incorrect and has now been corrected.

To maintain alignment between the supporting materials and the figures in the manuscript, the relevant images in the paper have also been updated.

2. Figures 4(C, D) have been adjusted and revised to harmonize the color schemes used in the bar chart with those in Figure B, thereby minimizing potential confusion.

3. Figure 5 has been revised to ensure that the colors in Figures G and H match those in the preceding figure.

4. This clarification has already been incorporated into the Materials and Methods section. It should be noted that Figures 4 and 2 represent two distinct experiments conducted using two separate batches of rats.

---

## [Editor Report · Decision Letter 2]

Dynamic CYP2E1 expression and metabolic activity changes in male rats during immune liver injury and sex differences in alcohol metabolism

PONE-D-25-09524R2

Dear Dr. xue,

We’re pleased to inform you that your manuscript has been judged scientifically suitable for publication and will be formally accepted for publication once it meets all outstanding technical requirements.

Kind regards,

Matias A Avila, Ph.D.

Academic Editor

PLOS ONE
---

## [Editor Report · Acceptance letter]

PONE-D-25-09524R2

PLOS ONE

Dear Dr. xue,

I'm pleased to inform you that your manuscript has been deemed suitable for publication in PLOS ONE. Congratulations! Your manuscript is now being handed over to our production team.

Kind regards,

on behalf of

Dr Matias A Avila

Academic Editor

PLOS ONE